# SARS-CoV-2 Infection and C-Section: A Prospective Observational Study

**DOI:** 10.3390/v13112330

**Published:** 2021-11-22

**Authors:** Eva Morán Antolín, José Román Broullón Molanes, María Luisa de la Cruz Conty, María Begoña Encinas Pardilla, María del Pilar Guadix Martín, José Antonio Sainz Bueno, Laura Forcén Acebal, Pilar Pintado Recarte, Ana Álvarez Bartolomé, Juan Pedro Martínez Cendán, Óscar Martínez-Pérez

**Affiliations:** 1Department of Gynecology and Obstetrics, Son Espases University Hospital, 07120 Palma de Mallorca, Spain; emoranantolin@yahoo.es; 2Department of Gynecology and Obstetrics, Puerta del Mar University Hospital, 1109 Cádiz, Spain; jrbroullon@gmail.com; 3Fundación de Investigación Biomédica, Puerta de Hierro University Hospital of Majadahonda, 28222 Madrid, Spain; farmcruz@gmail.com; 4Department of Gynecology and Obstetrics, Puerta de Hierro University Hospital of Majadahonda, 28222 Madrid, Spain; beenpar@yahoo.es; 5Department of Gynecology and Obstetrics, Virgen de la Macarena University Hospital, 41009 Sevilla, Spain; pilarguadix@gmail.com; 6Department of Gynecology and Obstetrics, G. Chacon (Viamed Santa Angela de la Cruz Hospital), 41014 Sevilla, Spain; jsainz@us.es; 7Department of Gynecology and Obstetrics, 12 de Octubre University Hospital, 28041 Madrid, Spain; lauratrona@gmail.com; 8Department of Gynecology and Obstetrics, Gregorio Marañon University Hospital, 28007 Madrid, Spain; ppintadorec@yahoo.es; 9Department of Anesthesia and Resuscitation, Puerta de Hierro University Hospital of Majadahonda, 28222 Madrid, Spain; anaalvarezbartolome@gmail.com; 10Department of Gynecology and Obstetrics, Santa Lucia University Hospital, 30202 Cartagena, Spain; jmartinezcen@gmail.com; 11Medical Simulation Department, Universidad Católica de Murcia, 30107 Murcia, Spain; 12Obstetric and Gynecology Department, Universidad Autónoma de Madrid, 28029 Madrid, Spain

**Keywords:** SARS-CoV-2, COVID-19, pregnancy, delivery, C-section, Robson’s ten group, perinatal outcomes, pneumonia

## Abstract

Pregnant women are particularly vulnerable to the severe acute respiratory syndrome coronavirus 2 (SARS-CoV-2) pandemic. In addition to unfavorable perinatal outcomes, there has been an increase in obstetric interventions. With this study, we aimed to clarify the reasons, using Robson’s classification model, and risk factors for cesarean section (C-section) in SARS-CoV-2-infected mothers and their perinatal results. This was a prospective observational study that was carried out in 79 hospitals (Spanish Obstetric Emergency Group) with a cohort of 1704 SARS-CoV-2 PCR-positive pregnant women that were registered consecutively between 26 February and 5 November 2020. The data from 1248 pregnant women who delivered vaginally (vaginal + operative vaginal) was compared with those from 456 (26.8%) who underwent a C-section. C-section patients were older with higher rates of comorbidities, in vitro fertilization and multiple pregnancies (*p* < 0.05) compared with women who delivered vaginally. Moreover, C-section risk was associated with the presence of pneumonia (*p* < 0.001) and 41.1% of C-sections in patients with pneumonia were preterm (Robson’s 10th category). However, delivery care was similar between asymptomatic and mild–moderate symptomatic patients (*p* = 0.228) and their predisposing factors to C-section were the presence of uterine scarring (due to a previous C-section) and the induction of labor or programmed C-section for unspecified obstetric reasons. On the other hand, higher rates of hemorrhagic events, hypertensive disorders and thrombotic events were observed in the C-section group (*p* < 0.001 for all three outcomes), as well as for ICU admission. These findings suggest that this type of delivery was associated with the mother’s clinical conditions that required a rapid and early termination of pregnancy.

## 1. Introduction

In March 2020, severe acute respiratory syndrome coronavirus 2 (SARS-CoV-2) was declared by the World Health Organization (WHO) to have caused a pandemic [1,2]. SARS-CoV-2 infects mainly through respiratory droplets, binds to the angiotensin converting enzyme 2 (ACE2) receptor and enters lung epithelial cells causing severe pathogenesis. Most people with a fully functional immune system who are exposed to SARS-CoV-2 undergo asymptomatic infection, while 5–10% are symptomatic and 1–2% are critically affected. These severely affected patients display a cytokine storm due to a dysfunctional immune response, which brutally destroys the affected organs and can lead to death [3]. Older people or people with comorbidities are high-risk population groups; however, pregnant women should also be included in this category, as the immunological and physiological changes of pregnancy make them more vulnerable to respiratory infections [4].

It is now known that SARS-CoV-2 infection presents a similar clinical picture in pregnant women to non-pregnant women, and only a small percentage of the former develop pneumonia due to coronavirus disease 2019 (COVID-19) (0–14%) and severe maternal and neonatal complications [5]. Hypertension, obesity, diabetes mellitus, previous cardiopulmonary diseases and older maternal age are among the risk factors that have been described to be associated with complicated COVID-19 disease in this population [2].

Regarding the mode of delivery, a high rate of cesarean sections (C-sections) was observed in patients infected with SARS-CoV-2 in the early phases of the pandemic [6,7,8,9]. A C-section is a mode of delivery through an open abdominal incision (laparotomy) and an incision in the uterus (hysterotomy), before the removal of the fetus begins; it requires anesthesia and follow-up car [10]. This high rate of C-sections early in the pandemic seemed to be associated with severe COVID-19 pathology and a lack of knowledge of this disease.

Most international obstetrics and gynecology guidelines state that vaginal delivery in SARS-CoV-2-infected patients is safe [11] and, when a C-section is performed, it should be based on obstetric indications. However, although the frequency of C-sections has decreased throughout the pandemic, the rates in SARS-CoV-2 infected mothers remain high (above 25%), and significantly higher than those registered in non-infected mothers [12,13].

Furthermore, the indiscriminate use of C-sections in delivery care is a global public health problem and it is not associated with a reduction in maternal or neonatal mortality [14]. In order to characterize the reasons underlying the high use of C-sections in different settings in a standardized manner, Robson’s international classification has been widely implemented [15,16]. This system determines the clinical data that are needed to classify C-sections in different groups, which allows for further comparison and identification of C-sections’ rate trends. It is a 10-group classification model that is based on four obstetric parameters: previous obstetric history (previous deliveries and C-sections), onset of labor (spontaneous, induced or elective C-section), gestational category (multiple pregnancy or singleton pregnancy, with cephalic, breech or transverse presentation) and gestational age (in labor). 

Because of the existing doubt about a possible association of C-section with a subsequent worsening of the mother’s condition in patients infected with SARS-CoV-2, as described early in the pandemic [17], the present study was proposed. This study included pregnant women that were infected with SARS-CoV-2 during the three high-incidence waves of the pandemic in Spain (26 February 2020 to 30 April 2021), where the objective was to define the characteristics of the mothers who needed a C-section and to investigate the reasons using Robson’s classification model, risk factors for C-section in infected mothers and their perinatal results. 

## 2. Materials and Methods

### 2.1. Study Design

Here we present a multicenter prospective study, where consecutive cases of SARS-CoV-2 infection in a pregnancy cohort that were registered by 79 Spanish hospitals (members of the Spanish Obstetric Emergency Group, listed in Appendix A) were analyzed. The study procedures were approved by the Drug Research and Clinical Research Ethics Committee of Puerta de Hierro University Hospital (Madrid, Spain) on 23 March 2020 (protocol registration number, 55/20); afterward, each collaborating center obtained local protocol approval. The study protocol is available at ClinicalTrials.gov, identifier: NCT04558996. Informed consent was obtained from every mother that was willing to participate in the study. 

In order to record the information needed, a specific database was designed for the study and used by the lead researcher of each participating center. The data were entered after the delivery of each patient. STROBE guidelines for cohort studies were followed for the duration of the study (Appendix A) [18].

### 2.2. Study Participants

The recruitment took place between 26 February and 5 November 2020. Every pregnant woman that attended the participating hospitals and was diagnosed as positive for SARS-CoV-2 was included in the cohort. SARS-CoV-2 infection was diagnosed using positive double-sampling polymerase chain reaction (PCR) from nasopharyngeal swabs; this test was applied in suspicious cases that arrived at hospital due to compatible COVID-19 symptoms and to every woman at admission in the delivery ward (universal screening, which started on 1 April 2020), regardless of whether they had COVID-19 in the past. These patients were subsequently classified according to their gestational age at the diagnosis of SARS-CoV-2 infection (1st, 2nd or 3rd trimester of gestation), as well as their COVID-19 symptomatology upon diagnosis, i.e., asymptomatic or symptomatic, with the latter stratified into mild–moderate symptoms (cough, anosmia, fatigue/discomfort, fever, dyspnea, etc.) and pneumonia.

### 2.3. Recorded Information

The demographic characteristics, comorbidities and obstetric history of the study participants were extracted from their clinical history; afterward, the classification used by the CDC (Centers for Disease Control Prevention) was followed for age and race categorization [4]. We recorded the following perinatal events: the type of onset of labor and delivery, gestational age at delivery, preterm delivery (below 37 weeks), ICU admission and need for invasive mechanical ventilation, obstetrical complications (pre-eclampsia, hemorrhagic and thrombotic events) and maternal mortality. In addition, C-sections were characterized using Robson’s classification system [15,16]. On the other hand, neonatal data involved the five-minute Apgar score, umbilical artery pH, birth weight, admission to the neonatal intensive care unit (NICU) and neonatal mortality. The definition of the recorded clinical and obstetric conditions followed international criteria [14,19,20]. Patients were followed until six weeks postpartum; the last delivery registered in our database took place on 30 April 2021. Neonatal events were recorded until 14 days postpartum.

These infected patients were then divided into two groups based on the type of delivery: C-section vs. vaginal + operative vaginal delivery.

### 2.4. Statistical Analysis

Numerical variables were tested for a normal distribution with the Kolmogorov–Smirnov test. Medians and interquartile ranges (IQRs) were used for describing numerical variables; frequencies and percentages were used for categorical ones. The comparison between groups of interest was carried out using Mann–Whitney’s U test for numerical variables and Pearson’s chi-squared or Fisher´s exact test for categorical variables. Statistical tests were two-sided and were performed with SPSS V.20 (IBM Inc., Chicago, IL, USA); a *p*-value below 0.05 was considered statistically significant. 

In addition, multivariable logistic regression modeling was conducted in order to derive the adjusted odds ratio (aOR) with a 95% confidence interval (95% CI) of “C-section risks factors” found in the previous univariable analysis. The regression analysis was carried out using the Ime4 package in R, version 3.4 (RCoreTeam, 2017) [21].

## 3. Results

### 3.1. Results for the Entire SARS-CoV-2-Infected Cohort

#### 3.1.1. General Data

During the study period, a total of 1704 pregnant women with SARS-CoV-2 infection were diagnosed, either because of suspicious symptoms or during admission to the delivery room.Of the 1704 SARS-CoV-2 positive women, 26.8% (456/1704) underwent a C-section and 73.2% (1248/1704) delivered vaginally, either via vaginal (1071, 62.9%) or operative vaginal (177, 10.4%) deliveries (Figure 1).

#### 3.1.2. Baseline Characteristics, Maternal Comorbidities and Pregnancy Characteristics

The maternal age of the women who underwent a C-section was statistically higher than the group of women who delivered vaginally (*p* < 0.001); of the former, up to 44.3% were older than 35 years (Table 1).A higher proportion of nulliparous and smokers was observed in the C-section group (Table 1).A higher proportion of pregnant women with comorbidities (obesity, thrombophilia, chronic kidney disease and diabetes mellitus) was observed among those who underwent a C-section (Table 1).There were significantly more IVF and multiple pregnancies observed in the C-section group (Table 1), in addition to more cases of intrauterine growth restrictions and gestational hypertension (*p* = 0.002 and *p* < 0.001, respectively).A total of 8.2% of women in the C-section group were classified as high risk for pre-eclampsia (by screening at 11–14 weeks of gestation) compared to 5.3% of women who delivered vaginally (*p* = 0.038, Table 1).

#### 3.1.3. Gestational Age at the Moment of SARS-CoV-2 Infection Diagnosis

Of the 1704 SARS-CoV-2-positive women in our cohort, 92 (5.4%) were diagnosed with SARS-CoV-2 infection in the first trimester of gestation, 292 (17.1%) in the second trimester and 1320 (77.5%) in the third trimester.The rates of C-sections among patients who were diagnosed in the first, second and third trimester of gestation were 16.35% (15/92), 24.7% (72/292) and 28.0% (369/1320), respectively, with the risk of a cesarean section being significantly higher when the infection took place late in pregnancy (*p* = 0.034).

#### 3.1.4. Maternal and Neonatal Outcomes

Gestational age at delivery was significantly lower among women who underwent a C-section (*p* < 0.001, Table 2), with higher rates of preterm delivery (<37 weeks of gestational age) in this group (23.5% vs. 6.3% of patients who delivered vaginally, *p* < 0.001).A higher incidence of obstetric and medical complications (hemorrhagic events, hypertensive disorders and thrombotic events) was observed among patients who underwent a C-section (Table 2), as well as of ICU admissions and requirement of invasive mechanical ventilation (Table 2).There were two cases of maternal death among the patients of the C-section group, both of which were associated with disseminated intravascular coagulation, and none in the vaginal delivery group (*p* = 0.071).Higher rates of newborns with low Apgar scores, low umbilical artery pH and NICU admissions were observed in the C-section group (Table 2).

### 3.2. Results for the Third Trimester Infections

#### 3.2.1. COVID-19 Symptomatology and Delivery Characteristics

More than a third of the patients had labor induced regardless of COVID-19 symptomatology (Table 3) and only 38.4% of patients who developed pneumonia had a spontaneous onset.The type of delivery varied according to COVID-19 symptomatology: the proportions of C-sections among the asymptomatic patients, the ones who had COVID-19 mild–moderate symptoms and the ones who developed pneumonia were 23.5, 28.1 and 40.8%, respectively (*p* < 0.001, Table 3).Among the patients who underwent a C-section (a total of 369), 38 (10.3%) were admitted to the ICU, whereas only 0.6% of patients who delivered vaginally (6/951) needed intensive care (*p* < 0.001).Regarding Robson’s classification of C-sections, there was a higher proportion of patients belonging to the 4th (multiparous without previous C-section, singleton pregnancy with cephalic presentation, ≥37 weeks’ gestation) and the 10th (singleton pregnancy with cephalic presentation, <37 weeks’ gestation, including those who had one or more previous C-section) categories among those who developed pneumonia (Table 3).Of the patients who developed pneumonia, underwent a C-section and belonged to Robson’s fourth category, 77.8% had an induced labor and 22.2% had a programmed C-section.Only 17.9% (17/95) of patients with pneumonia and who underwent a C-section had a spontaneous onset of labor.The highest proportion of asymptomatic patients who underwent a C-section belonged to Robson’s second category (24.7%: nulliparous women, singleton pregnancy with cephalic presentation, ≥37 weeks’ gestation, induced labor or programmed C-section) and mild–moderate symptomatic patients to the fifth category (22.5%: multiparous women with previous C-section, singleton pregnancy with cephalic presentation, ≥37 weeks’ gestation, spontaneous onset of labor).Nearly one-third of patients who developed pneumonia and underwent a C-section were admitted to the ICU (30/95, 31.6%); from these, 66.7% (20/30) underwent a C-section before ICU admission and 33.3% (10/30) afterward (Table 3).

#### 3.2.2. Description of C-Sections by Gestational Age at Delivery

Nearly 25% (89/369) of all C-sections (registered in mothers that were infected with SARS-CoV-2 in the third trimester of gestation) took place before the 37th week of gestation; of these, up to 76.4% (68/89) belonged to Robson’s 10th category (singletons pregnancies < 37 weeks with cephalic presentation, with or without previous C-sections).The C-section rate decreased as gestational age at delivery increased (and vice versa) (Table 4, *p* < 0.001).The main reason for C-section in very early preterms (<33 weeks of gestation) was COVID-19 worsening or complication and, as gestational age advanced, these were due to obstetric conditions (Table 4, *p* < 0.001).Up to 25% (70/279) of C-sections at term were due to induction failure or programmed C-section in nulliparous women (Robson’s second category) and nearly 20% (51/279) were due to the same reasons but in multiparous women (Robson’s fourth category).

### 3.3. Multivariable Analysis of Risk Factors for Undergoing a C-Section


The multivariable logistic regression modeling results (Table 5) corroborated that the following conditions significantly increased the risk of C-section in SARS-CoV-2-infected mothers: being an IVF pregnancy, being diagnosed with a SARS-CoV-2 infection in the third trimester of gestation, prematurity in mothers with COVID-19 pneumonia (although both conditions, by themselves, were risk factors for a C-section) and developing pre-eclampsia.


## 4. Discussion

Our study provides information on the characteristics of SARS-CoV-2-infected mothers according to the type of delivery, the causes and risk factors for C-section and their perinatal results. The main strength of the study was the large cohort of SARS-CoV-2-positive deliveries (1704), the participation of different hospitals (79 centers, public and private) across Spain and the long duration of the study (deliveries that took place from 26 February 2020 to 30 April 2021). In addition, the reasons for C-sections were standardly characterized with Robson’s classification system, taking into consideration the possible differences in clinical practice of the several participating hospitals.

The global incidence of C-sections in our SARS-CoV-2 infected cohort was 26.8%, lower than the one reported by previous studies [7,8]. This difference may be related to the PCR universal screening that was established in the participating hospitals, regardless of the mother’s symptomatology. Moreover, the long period of data collection may have resulted in a better understanding of the disease and, consequently, a decrease in the rate of C-sections. To this day, vertical transmission has not been demonstrated and vaginal delivery was shown to be safe in this COVID-19 scenario [11].

Mothers in the C-section group were older, which could be associated with comorbidities and infertility and, therefore, with a greater need for IVF, as shown by our results. COVID-19 has shown a more aggressive course in patients with previous comorbidities due to its systemic manifestations, such as hypertension, renal disease, thrombocytopenia and liver damage [12,22]. Therefore, the more severe the COVID-19 is, the more likely it is that the pregnancy will end via C-section, especially in women with pneumonia. Furthermore, and prior to the COVID-19 pandemic, IVF and multiple pregnancies had already been described as risk factors for C-section and obstetric morbidity [23,24,25].

In addition, there was a higher proportion of mothers with obesity, diabetes mellitus, hypertension, smoking and multiple gestations in the C-section group. Patients with these characteristics present more obstetric complications [22], which are associated with placental abnormalities and coagulation disorders. These situations could be aggravated by the SARS-CoV-2 infection [26].

Regarding the association that was observed between the moment of SARS-CoV-2 diagnosis and the type of delivery, the risk of C-section was significantly higher when the infection took place late in pregnancy. This fact could be related to maternal and/or obstetrician’s preferences regarding the uncertainty resulting from a PCR positive result (especially at the beginning of the pandemic) and, of course, with the presence of COVID-19 pneumonia in the third trimester of pregnancy. Furthermore, mothers that were diagnosed with SARS-CoV-2 infection during their first trimester delivered with past infection and late in the pandemic and, by then, hospitals already had experience in managing delivery in a COVID-19 scenario.

When it came to perinatal outcomes, the association that was observed between C-section and preterm delivery, as well as with ICU admission (and the need for invasive mechanical ventilation), could be explained by the urgency to terminate the pregnancy due to a worsening of the mother’s condition. If we compare the type of delivery according to COVID-19 symptomatology of the mother, a significant proportion of C-sections occurred in the pneumonia group, which seems to be justified by the respiratory failure of these patients, whereas asymptomatic patients had similar C-section rates to those with mild–moderate COVID-19 symptoms. Moreover, we must always bear in mind that maternal oxygen consumption increases 20% during pregnancy owing to increased metabolic demands and this, combined with reduced functional residual capacity, results in rapid desaturation during respiratory compromise [27], and it is at this moment when intervention is needed. In two-thirds of our patients with pneumonia who underwent a C-section, pregnancy was terminated before ICU admission, and in one-third following. Contrary to what our group reported in the first months of the COVID-19 pandemic with a small series of patients [17], it does not seem that C-section was the determining cause of ICU admission but rather that obstetricians decided on this type of delivery just before the patient with pneumonia was admitted to the ICU. Therefore, this type of delivery might be a consequence of the mother’s worsening condition and not a risk factor. It is important to note that many COVID-19 patients in the ICU require decubitus prone positions to improve pulmonary perfusion, where this is especially complicated in pregnant patients. 

Previous studies already described the association between SARS-CoV-2 pneumonia and iatrogenic prematurity [12,13,17]; this, although inevitable given the maternal deterioration, results in increased neonatal morbidity. 

Additionally, patients in the C-section group had more obstetric and medical complications (abruptio placentae, hypertensive disorders and thrombotic events), including two maternal deaths that were associated with disseminated intravascular coagulation. It was established that women with a SARS-CoV-2 infection have a significantly higher risk of developing pre-eclampsia [28]; this results in more C-sections in preterm pregnancies due to the difficulty in inducing these deliveries, which is in line with the findings mentioned above. 

Still, a C-section may result in postpartum hemorrhage and postpartum thrombotic events [29,30], as shown in Table 2; therefore, an individual risk–benefit assessment should be made before a C-section is performed. 

Regarding the Robson’s classification results, the reasons for the higher rate of C-sections among patients with pneumonia were preterm births (10th category) and at term onset of labor via induction or programmed C-section (4th category). In preterm births, the mother’s worsening condition would explain the urgency to terminate the pregnancy. On the other hand, nearly 45% of C-sections at term, regardless of the parity and the clinical presentation of SARS-CoV-2 infection, were due to induction failure or programmed C-section. The rate of C-sections in the group of asymptomatic, nulliparous and at term pregnant women (second category) reflected an increase of inductions and programmed C-sections that cannot be explained by the associated morbidities described above. Moreover, the contribution of multiparous women with a previous C-section (fifth category) and COVID-19 mild–moderate symptomatology to this C-section rate was likely related to an aversion to attempting vaginal deliveries. It is necessary to remember that the first Chinese series reported unusually high rates of C-sections and was surely influenced by the initial obstetric strategies during the first wave of the pandemic [31].

Among the limitations of the study is the overrepresentation of symptomatic infections in our cohort because not all participating hospitals had a universal antenatal screening program for SARS-CoV-2 and only identified symptomatic cases via passive surveillance, or implemented this screening program later. On the other hand, the equipment used for the PCR technique differed between hospitals, but all were based on similar extraction and amplification principles. In addition, no serological test was performed on most of our patients to confirm their disease and immune response, either because the tests were not available at the time of recruitment or because of logistical constraints due to the health sector crisis. Lastly, obstetricians’ experience and guidelines followed regarding the delivery care of SARS-CoV-2-infected mothers (and especially regarding C-section use under these circumstances) may vary between centers; however, in the statistical analyses, it was not possible to control for hospitals; the Robson classification of C-sections was applied instead.

In conclusion, C-section risk in SARS-CoV-2-infected patients is associated with the presence of pneumonia, especially in preterm pregnancies, and with clinical conditions that require a rapid and early termination of pregnancy. In pregnancies at term, delivery care is similar between asymptomatic patients and those with mild–moderate COVID-19 symptoms such that the presence of uterine scarring (due to a previous C-section) and induction or programmed C-sections for unspecified obstetric reasons contributed to the C-section rate. Therefore, clinical practice guidelines to improve the quality of pregnancy care in times of COVID-19 are needed, as well as a standardization of the indications for C-sections. 

## 5. Conclusions

After studying the influence of SARS-CoV-2 infection on the type of delivery, we concluded that COVID-19 pneumonia, especially in preterm pregnancies and/or associated with comorbidities, increased the risk of C-section. In contrast, in infected pregnancies at term, the predisposing factors for C-section were the presence of uterine scarring (due to a previous C-section) and induction or programmed C-section for unspecified obstetric reasons.

## Figures and Tables

**Figure 1 viruses-13-02330-f001:**
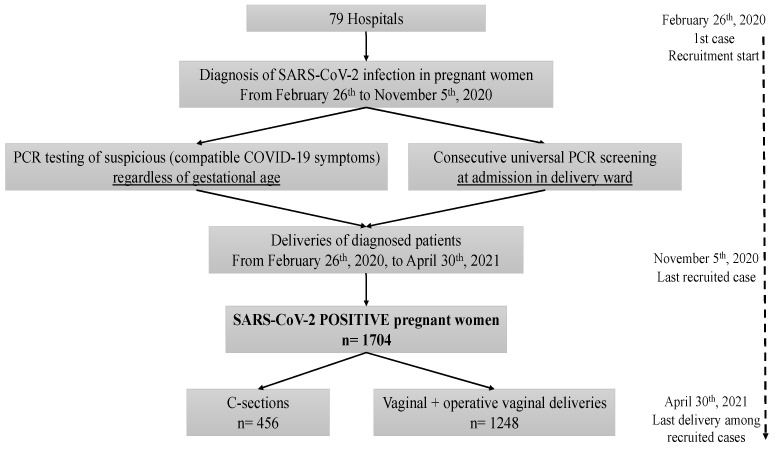
Flow chart of the study data.

**Table 1 viruses-13-02330-t001:** Demographic characteristics, comorbidities and current obstetric history of the study participants (n = 1704).

		Infected Cohort	Vaginal + Operative Vaginal	C-Section	*p*-Value
Number		1704	1248 (73.2)	456 (26.8)	
**Maternal Characteristics**					
Maternal age (years; median/IQR)		32 (28–36)	32 (27–36)	34 (29–37)	<0.001 *
Age Range	18–24 years	241/1689 (14.3) 825/1689 (48.8) 623/1689 (36.9)	188 (15.2) 627 (50.6) 423 (34.2)	53 (11.8) 198 (43.9) 200 (44.3)	0.001 *
	25–34 years				
	35–49 years				
Ethnicity	White European	947/1699 (55.7) 514/1699 (30.3) 44/1699 (2.6) 50/1699 (2.9) 144/1699 (8.5)	689/1245 (55.3) 374/1245 (30.0) 34/1245 (2.7) 39/1245 (3.1) 109/1245 (8.8)	258/454 (56.8) 140/454 (30.8) 10/454 (2.2) 11/454 (2.4) 35/454 (7.7)	0.816
	Latino Americans				
	Black non-Hispanic				
	Asian non-Hispanic				
	Arab				
Nulliparous		616/1688 (36.5)	431/1233 (35.0)	185/455 (40.7)	0.031 *
Smoking ^a^		160 (9.7)	105 (8.7)	55 (12.7)	0.016 *
**Maternal Comorbidities**					
Obesity (BMI > 30 kg/m^2^)		317 (18.6)	203 (16.3)	114 (25.0)	<0.001 *
Thrombophilia		28 (1.6)	15 (1.2)	13 (2.9)	0.018 *
Chronic kidney disease		6 (0.4)	2 (0.2)	4 (0.9)	0.047 *
Diabetes mellitus		36 (2.1)	20 (1.6)	16 (3.5)	0.015 *
**Current Obstetric History**					
Multiple pregnancy		31 (1.8)	12 (1.0)	19 (4.2)	<0.001 *
In vitro fertilization		82 (4.8)	37 (3.0)	45 (9.9)	<0.001 *
Intrauterine growth restriction		61 (3.6)	34 (2.7)	27 (5.9)	0.002 *
Pregnancy-induced hypertension ^b^		42 (2.5)	20 (1.6)	22 (4.8)	<0.001 *
High-risk pre-eclampsia screening		90/1484 (6.1)	58/1095 (5.3)	32/389 (8.2)	0.038 *

Data shown as n (% of total), except otherwise indicated. * Statistically significant differences. BMI: body mass index; HBP: high blood pressure. ^a^ Current smokers + ex-smokers. ^b^ Hypertension + pre-eclampsia.

**Table 2 viruses-13-02330-t002:** Maternal and neonatal outcomes of the study participants (n = 1704).

	Infected Cohort	Vaginal + Operative Vaginal	C-Section	*p*-Value
Number	1704	1248 (73.2)	456 (26.8)	
**Outcomes**				
Gestational age at delivery (weeks + days; median/IQR)	39 + 7 (38 + 2 to 40 + 3)	39 + 4 (38 + 4 to 40 + 1)	39 + 0 (37 + 0 to 40 + 1)	<0.001 *
Hemorrhagic events Abruptio placentae Postpartum hemorrhage	93 (5.5) 18 (1.1) 79 (4.6)	48 (3.8) 1 (0.1) 47 (3.8)	45 (9.9) 17 (3.7) 32 (7.0)	<0.001 * <0.001 * 0.005 *
Hypertensive disorders Antepartum/postpartum hypertension Pre-eclampsia/eclampsia Moderate pre-eclampsia Severe pre-eclampsia/HELLP/eclampsia	104/1661 (6.3) 87 (5.1) 56/87 (64.4) 31/87 (35.6)	50/1222 (4.1) 42 (3.4) 30/42 (71.4) 12/42 (28.6)	54/439 (12.3) 45 (9.9) 26/45 (57.8) 19/45 (42.2)	<0.001 * <0.001 * 0.184
Thrombotic events Deep venous thrombosis Pulmonary embolism Disseminated intravascular coagulation	18 (1.1) 3 (0.2) 10 (0.6) 6 (0.4)	6 (0.5) 1 (0.1) 3 (0.2) 2 (0.2)	12 (2.6) 2 (0.4) 7 (1.5) 4 (0.9)	<0.001 * 0.176 0.005 * 0.047 *
Admitted in ICU	52 (3.1)	7 (0.6)	45 (9.9)	<0.001 *
Invasive mechanical ventilation	31 (1.8)	3 (0.2)	28 (6.1)	<0.001 *
**Maternal Mortality**	2 (0.1)	0 (0.0)	2 (0.4)	0.071
**Neonatal Score**				
Apgar 5 score < 7 Umbilical artery pH < 7.10 Birth weight (grams; median/IQR)	17/1661 (1.0) 44/1359 (3.2) 3260 (2900–3570)	3/1218 (0.2) 25/989 (2.5) 3290 (2940–3560)	14/443 (3.2) 19/370 (5.1) 3170 (2628–3595)	<0.001 * 0.016 * <0.001 *
Admitted in NICU	163/1684 (9.7)	70/1234 (5.7)	93/450 (20.7)	<0.001 *
Neonatal mortality	6 (0.4)	2 (0.2)	4 (0.9)	0.047 *

Data shown as n (% of total), except otherwise indicated. * Statistically significant differences.

**Table 3 viruses-13-02330-t003:** Description of the onset of labor, mode of delivery and the reasons for C-section categorized by the clinical presentation of SARS-CoV-2 infection in patients infected in the 3rd trimester of gestation (n = 1320).

	Asymptomatic	Mild–Moderate Symptoms	Pneumonia	*p*-Value
Number	689 (52.2)	398 (30.2)	233 (17.7)	
Onset of labor: Programmed C-section Spontaneous Induced	46 (6.7) 392 (56.9) 251 (36.4)	46 (11.6) 199 (50.0) 153 (38.4)	54/232 (23.3) 89/232 (38.4) 89/232 (38.4)	<0.001 *
Type of delivery: Vaginal Operative vaginal C-section	450 (65.3) 77 (11.2) 162 (23.5)	242 (60.8) 44 (11.1) 112 (28.1) ^a^	119 (51.1) 19 (8.2) 95 (40.8)	<0.001 *^,b^
Robson classification of C-sections: 1 2 3 4 5 6 7 8 9 10	15/162 (9.3) 40/162 (24.7) 14/162 (8.6) 21/162 (13.0) 30/162 (18.5) 12/162 (7.4) 12/162 (7.4) 4/162 (2.5) 1/162 (0.6) 13/162 (8.0)	8/111 (7.2) 20/111 (18.0) 10/111 (9.0) 12/111 (10.8) 25/111 (22.5) 9/111 (8.1) 6/111 (5.4) 5/111 (4.5) 0/111 (0.0) 16/111 (14.4)	2/95 (2.1) 10/95 (10.5) 5/95 (5.3) 18/95 (18.9) 10/95 (10.5) 5/95 (5.3) 2/95 (2.1) 4/95 (4.2) 0/95 (0.0) 39/95 (41.1)	<0.001 *
C-section before or after ICU admission: Before ICU admission After ICU admission	2/162 (1.2) 2/2 (100) 0/2 (0.0)	6/112 (5.4) 4/6 (66.7) 2/6 (33.3)	30/95 (31.6) 20/30 (66.7) 10/30 (33.3)	0.614

Data shown as n (% of total). * Statistically significant differences. ^a^ One patient had missing data on the obstetric parameters that were needed for the Robson’s classification. ^b^ Difference due to pneumonias relative to the other two groups of patients; there was no statistically significant difference between asymptomatic patients and patients with mild–moderate symptoms (*p* = 0.228).

**Table 4 viruses-13-02330-t004:** Description of C-sections by gestational age range at delivery in mothers that were infected in the 3rd trimester of gestation (n = 1320).

	Gestational Age Range at Delivery				*p*-Value
	28 to <33 Weeks	33 to <37 Weeks	37 to <41 Weeks	≥41 Weeks	
Number	n = 39	n = 112	n = 986	n = 183	
Type of delivery: Vaginal + operative vaginal C-section	12 (30.8) 27 (69.2)	50 (44.6) 62 (55.4)	746 (75.7) 240 (24.3) ^a^	143 (78.1) 40 (21.9)	<0.001 *
Reasons for C-section: COVID-19 complication COVID-19 complication + pre-eclampsia COVID-19 complication + other obstetrical causes Pre-eclampsia without COVID-19 complication Other obstetrical causes without COVID-19 complication	11/27 (40.7) 4/27 (14.8) 2/27 (7.4) 4/27 (14.8) 6/27 (22.2)	19/62 (30.6) 4/62 (6.5) 7/62 (11.3) 5/62 (8.1) 27/62 (43.5)	25/240 (10.4) 3/240 (1.2) 17/240 (7.1) 15/240 (6.2) 180/240 (75.0)	5/40 (12.5) 0/40 (0.0) 0/40 (0.0) 1/40 (2.5) 34/40 (85.0)	<0.001 *
Robson classification of C-sections: 1 2 3 4 5 6 7 8 9 10	0/27 (0.0) 0/27 (0.0) 0/27 (0.0) 0/27 (0.0) 0/27 (0.0) 0/27 (0.0) 1/27 (3.7) 4/27 (14.8) 0/27 (0.0) 22/27 (81.5)	0/62 (0.0) 0/62 (0.0) 0/62 (0.0) 0/62 (0.0) 0/62 (0.0) 7/62 (11.3) 4/62 (6.5) 5/62 (8.1) 0/62 (0.0) 46/62 (74.2)	20/239 (8.4) 48/239 (20.1) 26/239 (10.9) 42/239 (17.6) 64/239 (26.8) 19/239 (7.9) 15/239 (6.3) 4/239 (1.7) 1/239 (0.4) 0/239 (0.0)	5/40 (12.5) 22/40 (55.0) 3/40 (7.5) 9/40 (22.5) 1/40 (2.5) 0/40 (0.0) 0/40 (0.0) 0/40 (0.0) 0/40 (0.0) 0/40 (0.0)	<0.001 *

Data shown as n (% of total). * Statistically significant differences. ^a^ One patient had missing data on the obstetric parameters that were needed for the Robson’s classification.

**Table 5 viruses-13-02330-t005:** Multivariable analysis of the C-section risk.

Multivariable Model	Variables Associated with C-Section	*p*-Value	aOR (95% CI)
C-section = COVID-19 symptoms + preterm delivery + interaction (COVID-19 symptoms and preterm delivery) + gestational age at diagnosis + pre-eclampsia + IVF	COVID-19 mild–moderate symptoms	0.523 ^a^	
	**COVID-19 pneumonia**	**0.013 ^a^**	**1.55 (1.09–2.18)**
	**Preterm delivery**	**0.003**	**2.44 (1.34–4.37)**
	Interaction (COVID-19 mild–moderate symptoms and preterm delivery)	0.456	
	**Interaction (COVID-19 pneumonia and preterm delivery)**	**0.013**	**2.99 (1.27–7.25)**
	Diagnosis in 2nd trimester	0.141 ^b^	
	**Diagnosis in 3rd trimester**	**0.029 ^b^**	**1.94 (1.10–3.64)**
	**Pre-eclampsia**	**<0.001**	**2.51 (1.55–4.04)**
	**IVF**	**<0.001**	**3.38 (2.10–5.44)**

COVID-19 symptoms: 3 categories—asymptomatic, mild-moderate symptoms and pneumonia. Gestational age at diagnosis: 3 categories—1st trimester, 2nd trimester and 3rd trimester. Pre-eclampsia: moderate + severe. IVF: own oocyte + donor oocyte. ^a^ Compared to basal category—asymptomatic. ^b^ Compared to basal category—1st trimester.

## Data Availability

The data presented in this study are available on request from the corresponding author. The data are not publicly available due to the multicenter nature of the study.

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
