# Peer review of "SARS-CoV-2 Infection and C-Section: A Prospective Observational Study"

_viruses, 2021, doi:10.3390/v13112330_

Round 1
Reviewer 1 Report
Thank you for the opportunity to review the Moran-Antolin manuscript entitled: SARS-CoV-2 infection and C-section: a prospective observational study. The therapeutic management of pregnant women infected with SARS-CoV-2 is still under research. Therefore, prospective studies of this problem are very important when the next wave of the pandemic is approaching.
I have a few minor comments on the manuscript.
In my opinion, unnecessary self-citations should be ruled out, as they do not add much to the text - references nr 13 and 33.
122-131 - If the diagnosis of SARS was made in the first, second and third trimesters, was it further confirmed on the day of delivery or confirmed by immunological tests?
132-146 - Did the patients have an immunological test performed 6 weeks after delivery? Have newborns been tested for SARS?
368-370 Is there any information what percentage of newborns were ill between natural delivery and caesarean section in these patients?
403-410 The fetus is also a patient of the obstetrician, therefore I miss in this manuscript, a reference to the obstetric results of newborns.
414-419 In the case of caesarean section, the absolute indications for the procedure are always the same. On the other hand, relative indications are subject to change. The manuscript conclusion is in line with the global trend of limiting cesarean sections, the conclusion is consistent with the results presented above.
Author Response
|
REVIEWER #1 |
AUTHOR´S RESPONSE AND CHANGES |
|
|
Thank you for the opportunity to review the Moran-Antolin manuscript entitled: SARS-CoV-2 infection and C-section: a prospective observational study. The therapeutic management of pregnant women infected with SARS-CoV-2 is still under research. Therefore, prospective studies of this problem are very important when the next wave of the pandemic is approaching.
I have a few minor comments on the manuscript: |
Thank you very much for your review and comments. |
|
|
1 |
In my opinion, unnecessary self-citations should be ruled out, as they do not add much to the text - references nr 13 and 33. |
Thank you; as suggested, references have been removed throughout the manuscript, except JAMA’s one because it is necessary in the hypothesis and discussion (when referring to “our group”). |
|
2 |
122-131 - If the diagnosis of SARS was made in the first, second and third trimesters, was it further confirmed on the day of delivery or confirmed by immunological tests? |
Thank you for the comment. Every pregnancy in the delivery ward was screened for SARS-CoV-2 infection. Women diagnosed before this moment (first, second and third trimester pregnancies before delivery) was because these patients came into hospital due to compatible COVID-19 symptoms. However, and regardless of whether or not they had COVID-19 in the past, these patients underwent a PCR test in the delivery ward. We have clarified this point in the Materials and Methods section (line 159, revised highlighted manuscript). On the other hand, in March 2020, when our study began, and during the first wave, there were not enough immunological tests for all these patients. Later, some hospitals used immunological tests in parallel to PCR, but it was the result from this second test the one that was considered for infection diagnosis. Finally, and due to a possible bias resulting from past infections results, only patients infected in the third trimester, near delivery, were taken into account for some of the analyses carried out in the study (Tables 3 and onwards). |
|
3 |
132-146 - Did the patients have an immunological test performed 6 weeks after delivery? Have newborns been tested for SARS? |
Thank you for this appreciation. As mentioned above, when our study began there were not enough immunological tests for all these patients, therefore, no immunological tests were performed 6 weeks after delivery. This point has been added as a limitation of our study (lines 462-465, revised highlighted manuscript). On the other hand, newborns underwent nasopharyngeal PCR tests within 12 hours after delivery during the first months of the pandemic; five positive newborns were identified and these cases were described in the following published article: Mejía Jiménez I, Salvador López R, García Rosas E, Rodriguez de la Torre I, Montes García J, de la Cruz Conty ML, Martínez Pérez O; Spanish Obstetric Emergency Group†. Umbilical cord clamping and skin-to-skin contact in deliveries from women positive for SARS-CoV-2: a prospective observational study. BJOG. 2021 Apr;128(5):908-915. doi: 10.1111/1471-0528.16597. Epub 2020 Nov 30. PMID: 33187026; PMCID: PMC7753553. All these positive newborns reported within 12 hours after delivery, tested negative in the confirmation test performed between 12 and 48 hours post-delivery. In addition, two other newborns developed COVID-19 symptoms within 10 days. Though initial testing at birth was negative, repeat testing was positive, which was possibly related with horizontal transmission through contact with a relative without the use of protection measures (and unknown infection). Both cases were described in another published article: Martínez-Perez O, Vouga M, Cruz Melguizo S, Forcen Acebal L, Panchaud A, Muñoz-Chápuli M, Baud D. Association Between Mode of Delivery Among Pregnant Women With COVID-19 and Maternal and Neonatal Outcomes in Spain. JAMA. 2020 Jul 21;324(3):296-299. doi: 10.1001/jama.2020.10125. Erratum in: JAMA. 2020 Jul 21;324(3):305. PMID: 32511673; PMCID: PMC7281380. Afterwards, PCR tests in newborns were stopped because, in many cases, the results were biased by horizontal contamination. |
|
4 |
368-370 Is there any information what percentage of newborns were ill between natural delivery and caesarean section in these patients? |
Thank you for the comment. As mentioned above, this information was covered by two published articles: 1. Mejía Jiménez I, Salvador López R, García Rosas E, Rodriguez de la Torre I, Montes García J, de la Cruz Conty ML, Martínez Pérez O; Spanish Obstetric Emergency Group†. Umbilical cord clamping and skin-to-skin contact in deliveries from women positive for SARS-CoV-2: a prospective observational study. BJOG. 2021 Apr;128(5):908-915. doi: 10.1111/1471-0528.16597. Epub 2020 Nov 30. PMID: 33187026; PMCID: PMC7753553. 2. Martínez-Perez O, Vouga M, Cruz Melguizo S, Forcen Acebal L, Panchaud A, Muñoz-Chápuli M, Baud D. Association Between Mode of Delivery Among Pregnant Women With COVID-19 and Maternal and Neonatal Outcomes in Spain. JAMA. 2020 Jul 21;324(3):296-299. doi: 10.1001/jama.2020.10125. Erratum in: JAMA. 2020 Jul 21;324(3):305. PMID: 32511673; PMCID: PMC7281380. Two out of these five newborns SARS-CoV-2 positive within 12 hours after delivery, and negative in the confirmation test, were born by C-section and 3/5 by vaginal or operative vaginal delivery. Even so, we would like to emphasize that the aim of the present study was to analyze the type of delivery in SARS-CoV-2 infected women, not these results in neonates (which have already been described by other studies), and that is the reason for not including this information in the present study. |
|
5 |
403-410 The fetus is also a patient of the obstetrician, therefore I miss in this manuscript, a reference to the obstetric results of newborns. |
Thank you for the comment. Results of newborns (Apgar 5 score, umbilical artery pH, birth weight, NICU admission and neonatal mortality) are shown in Table 2 and lines 278-279 (revised highlighted manuscript); these results were worse in the group born by C-section which, as described in our manuscript (lines 426-428 in the discussion section, revised highlighted manuscript) and previous ones (*), is related to iatrogenic prematurity resulting from the urgency to terminate the pregnancy due to a worsening of the mother’s condition. Although, as mentioned above, this was not the objective of the study, if the reviewer considers any other outcome to be noteworthy, it could be included in Table 2.
* We have also added the following references to discussion section:
|
|
6 |
414-419 In the case of caesarean section, the absolute indications for the procedure are always the same. On the other hand, relative indications are subject to change. The manuscript conclusion is in line with the global trend of limiting cesarean sections, the conclusion is consistent with the results presented above. |
Thank you very much for this review. |
Please see the attachment.

Reviewer 2 Report
The manuscript entitled “SARS-CoV-2 infection and C-section: a prospective observational study.” by Dr. Antolín and colleagues reports on a multicenter prospective observational study, aimed to identify reasons/risk factors for C-section in SARS-CoV-2 infected mothers, and carried out in 79 hospitals (Spanish Obstetric Emergency Group) with a cohort of 1,704 SARS-CoV-2 PCR-positive pregnant women registered consecutively between February 26th and November 5th, 2020. The main findings suggest that COVID-19 pneumonia, especially in preterm pregnancies and /or associated with comorbidities increases the risk of C-section, while in infected pregnancies at term, the predisposing factors to C-section would be the presence of uterine scarring and the induction or programmed C-section for unspecified obstetric reasons.
This work is relevant to the field, as it increases our knowledge on SARS-CoV-2 infection during pregnancy and the causes and risk factors for C-section. However, in my opinion, the ms lack in scientific saundness, while text should be reorganized, especially the results section. I have a several observations, and I am therefore recommending a major revision.
Thank you for letting me work as reviewer for this work
General comments
1. A brief explanation of C-section would be helpful for a non-expert reader, for instance “Cesarean section (C-section)”
2. Please remove spaces between paragraphs throughout the text. For instance lines 65 and 67, lines 72 and 75 , 337 and 338 etc..
3. COVID-19 etiopathogenesis should be, at least briefly, described in the introduction
4. Methods should be subdivided in subsections according to the topic. For instance, Pregnant females, samples collection, molecular analyses, statistics
5. Results should be described as a paragraph, without listing the main findings
6. Both RNA isolation and SARS-CoV-2-PCR detection procedure should be detailed in the methods section in a separated paragraph
Minor
Lines 40 and 66 SARS-CoV-2 and COVID-19 should be Severe acute respiratory syndrome due to coronavirus 2 (SARS-CoV-2) and Coronavirus disease 2019 (COVID-19), respectively, when mentioned for the fist time
Line 40 SARS-CoV-2
Line 63 Detailed information regarding the role of SARS-COv-2 in inducing COVID-19 is reported here (PMID: 34578269). This supporting reference should be included
Line 64 as playing a key role in viral infection establishment during pregnancy, immunological changes should also be mentioned, as described PMID: 17353679. The transient immune modulation occurring during pregnancy make pregnant females more susceptible to viral infections.
Lines 121 and 312 please revise the parenthesis
Line 214 The subhead 3.1.1. has been mistakenly listed. Please correct
Author Response
|
REVIEWER #2 |
AUTHOR´S RESPONSE AND CHANGES |
|
|
The manuscript entitled “SARS-CoV-2 infection and C-section: a prospective observational study.” by Dr. Antolín and colleagues reports on a multicenter prospective observational study, aimed to identify reasons/risk factors for C-section in SARS-CoV-2 infected mothers, and carried out in 79 hospitals (Spanish Obstetric Emergency Group) with a cohort of 1,704 SARS-CoV-2 PCR-positive pregnant women registered consecutively between February 26th and November 5th, 2020. The main findings suggest that COVID-19 pneumonia, especially in preterm pregnancies and /or associated with comorbidities increases the risk of C-section, while in infected pregnancies at term, the predisposing factors to C-section would be the presence of uterine scarring and the induction or programmed C-section for unspecified obstetric reasons.
This work is relevant to the field, as it increases our knowledge on SARS-CoV-2 infection during pregnancy and the causes and risk factors for C-section. However, in my opinion, the ms lack in scientific saundness, while text should be reorganized, especially the results section. I have a several observations, and I am therefore recommending a major revision.
Thank you for letting me work as reviewer for this work. |
Thank you for your comments; we have tried to address each one of them with detail. |
|
|
General comments: |
|
|
|
1 |
A brief explanation of C-section would be helpful for a non-expert reader, for instance “Cesarean section (C-section)” |
Thank you very much for this appreciation; we have added “Cesarean section (C-section)” when mentioned for the first time (line 86, revised highlighted manuscript) and have added a brief definition to the introduction section (lines 88-89, revised highlighted manuscript). In addition, we have extended the hypothesis paragraph (lines 111-113, revised highlighted manuscript). |
|
2 |
Please remove spaces between paragraphs throughout the text. For instance lines 65 and 67, lines 72 and 75 , 337 and 338 etc. |
Forgive us but we do not understand the request of removing the spaces between paragraphs; we believe that the spaces between paragraphs facilitate the reading of the manuscript, as well as that these spaces highlight when a paragraph begins and ends (in order to avoid confusion). |
|
3 |
COVID-19 etiopathogenesis should be, at least briefly, described in the introduction |
Thank you for this appreciation. This information has been added to the introduction section as suggested (lines 64-71, revised highlighted manuscript). |
|
4 |
Methods should be subdivided in subsections according to the topic. For instance, Pregnant females, samples collection, molecular analyses, statistics |
Thank you for this comment. We have divided the Materials and Methods section into subsections as suggested (lines 121, 152, 170, and 198, revised highlighted manuscript). |
|
5 |
Results should be described as a paragraph, without listing the main findings |
Forgive us but we do not agree with the reviewer on this point; we have tried to be concise and precise as it is stated in the instructions for authors of the journal, as well as we have followed the same schema as with previous articles submitted and published with this journal. Forgive us but we disagree with the reviewer on this point; we have tried to be concise and precise as indicated in the instructions for authors of the journal; moreover, we have followed the same scheme as with previous articles (submitted and published by this journal). |
|
6 |
Both RNA isolation and SARS-CoV-2-PCR detection procedure should be detailed in the methods section in a separated paragraph |
Thank you for this comment. The PCR technique used was mainly REAL-TIME PCR, using different equipment from different laboratories, as a total of 79 different hospitals (some large hospitals, some smaller) from all over Spain participated in the study, but all of them were based on similar extraction and amplification principles. Most of them were automated: high capacity equipment capable of processing a high number of samples in a short time; however, in the case of small or regional hospitals, or in the case of failure of the automated equipment, PCRs were performed manually by specialized technical personnel, or by rapid PCR equipment, such as the Roche cobas® liat®. In all cases, basic safety and hygienic measures were taken to avoid contagion and minimize contamination. Due to the heterogeneity resulting from the participation of 79 different hospitals, we consider that it is not possible a detailed description of this methodology; however, we have added this point as a limitation of the study (lines 460-462, revised highlighted manuscript). |
|
Minor comments: |
|
|
|
1 |
Lines 40 and 66 SARS-CoV-2 and COVID-19 should be Severe acute respiratory syndrome due to coronavirus 2 (SARS-CoV-2) and Coronavirus disease 2019 (COVID-19), respectively, when mentioned for the first time |
Thank you for this appreciation; corrected as suggested (lines 40 and 79, revised highlighted manuscript). |
|
2 |
Line 40 SARS-CoV-2 |
Corrected as suggested. |
|
3 |
Line 63 Detailed information regarding the role of SARS-COv-2 in inducing COVID-19 is reported here (PMID: 34578269). This supporting reference should be included |
Thank you for this suggestion; we have added this reference (line 63, revised highlighted manuscript). |
|
4 |
Line 64 as playing a key role in viral infection establishment during pregnancy, immunological changes should also be mentioned, as described PMID: 17353679. The transient immune modulation occurring during pregnancy make pregnant females more susceptible to viral infections. |
Thank you for this appreciation; we have clarified this point and have added the suggested reference (lines 71-73, revised highlighted manuscript). |
|
5 |
Lines 121 and 312 please revise the parenthesis |
Thank you; we have corrected those parentheses (line 140 and line 362, revised highlighted manuscript). |
|
6 |
Line 214 The subhead 3.1.1. has been mistakenly listed. Please correct |
Thank you; corrected as suggested (line 260, revised highlighted manuscript). |

Round 2
Reviewer 2 Report
The authors have addressed all my concerns and therefore I support publication.